# Hydrodynamic modeling of coastal seas: the role of tidal dynamics in the Messina Strait, Western Mediterranean Sea

A. Cucco<sup>\*</sup>, G. Quattrocchi, A. Olita, L. Fazioli, A. Ribotti, M. Sinerchia, C. Tedesco and R. Sorgente

National Research Council, Institute for Coastal Marine Environment, Oristano, 09170 Italy

\**Correspondence to*: A. Cucco (andrea.cucco@cnr.it)

**Abstract.** This work explored the importance of considering tidal dynamics when modeling the general circulation in the Messina Strait, a narrow passage connecting the Tyrrhenian and the Ionian Sea sub-basins in the Western Mediterranean Sea. The tides and the induced water circulation in this Strait are among the most intense oceanographic processes in the

- Mediterranean Sea. The quantification of these effects can be particularly relevant for operational oceanographic systems aimed to provide short-term predictions of the main hydrodynamics in the Western Mediterranean sub-basins. A numerical approach based on the use of a high resolution hydrodynamic model was adopted to firstly reproduce both the tides propagation and the wind induced and thermohaline water circulation within the Strait and surrounding areas and secondly to quantify the role of the Strait dynamics on the outer water circulation. The obtained results confirmed the importance of a
- correct representation of the hydrodynamics in the Messina Strait even when focusing on predicting the water circulation in the external sea traits. In fact, model results show that tidal dynamics deeply impact the reproduction of the instantaneous and residual circulation pattern, waters thermohaline properties and transport dynamics both inside the Messina Strait and in the surrounding coastal and open waters.

#### **1** Introduction

- In the XII chapter of the Odyssey, before the landing to Trinacria Island, Ulysses and his crew leaving the Circe refuge, experienced the wrath of Scilla and Cariddi, with great loss of men and ships (Homer, VI B.C.). The Homer's poem, describing the intense vortices and heavy currents generated by the tides (Scilla and Cariddi) in the Messina Strait (Western Mediterranean Sea, hereafter MS), can be considered one of the first examples of grey literature in physical Oceanography. The tides and the induced water circulation in this Strait (figure 1a and figure 1b) are among the most interesting
- oceanographic processes in the Mediterranean Sea, and not only because of Homer's epic. The intense current speeds and the high variability of tidal phases and frequencies lead to consider this area as one of the most energetic in terms of momentum and impulse all over the basin (Hopkins et al., 1984). This is why, in recent years,

several research activities were carried out to investigate how tidal dynamics in this area can be exploited to produce renewable energy (Coiro et al., 2013).

Even if notorious, the dynamic of the Strait is not fully addressed in scientific literatures, with only few and old studies describing the water circulation in both theoretical and experimental terms (Hopkins et al., 1984; Cescon et al., 1997) and

- very few recent investigations using numerical modelling techniques (Androsov et al., 2002a; Androsov et al., 2002b). In particular, while tidal dynamic inside the Strait has been studied and described by many authors starting from the early XX century (Vercelli, 1925; Vercelli, 1926;Defant, 1940; Bossolasco and Dagnino, 1957; Defant, 1961; Massi and Sallusti, 1979; Mosetti 1988), the effects of the Messina tidal in and outflow on the outer open ocean thermohaline water circulation is still unaddressed in scientific literature. In particular, both recent and old studies focused mainly on describing the
- behaviour of Tyrrhenian and Ionian waters flowing through the Strait (Bossolasco and Dagnino, 1959; Androsov et al., 2002b) and on the generation of internal waves (Brandt et al., 1997; Brandt and Rubino, 1999) without quantifying the role of MS tidal dynamics in modifying the outer circulation pattern.

The quantification of these effects can be particularly relevant for operational oceanographic systems aimed to provide short term predictions of the main hydrodynamics in the Western Mediterranean sub-basins. Most of this ocean prediction systems

- (Tonani et al., 2008; Oddo et al., 2009; Pinardi et al., 2010; Sorgente et al., 2011; Tonani et al., 2015) are not suitable to accurately reproduce the Strait dynamics mainly due to numerical grid limitation, where orthogonality and spatial resolution are not appropriate to describe fine scale coastal features. As a consequence, the operational output of such systems provide temperature, salinity and water currents fields, which are estimated ignoring the contribution of tidal exchanges within the Strait capable of modifying the water current, salinity and temperature fields as well as the water mass budgets between the
- sub-basins.

Consequently, the question is: how big is the effect accurately capturing finer scale processes, such as tidal dynamics, on the model reproduction of the circulation in the area? This issue is particularly relevant in the case of MS, that is characterized by intense tidal dynamics, quite a unique case in the Mediterranean sea, where tides are generally weak and have a low influence on circulation (Sannino et al., 2015).

- In this work, a numerical approach based on the use of a high resolution hydrodynamic model based on finite elements method was proposed to reproduce both the tides propagation and the wind induced and thermohaline circulation in the Strait and surrounding areas and to quantify the role of the Strait dynamics on the outer water circulation. Three different scenarios characterized by different model forcings were investigated in order to indentify the weight of each single contribution (tides, thermohaline and wind) to the main hydrodynamics in the area of interest.
- The paper is organized as follows: a brief description of the MS study area, including the morphological and oceanographic features is reported in Section 2. An overview of the applied method including the description of the adopted numerical model and of the three simulated scenarios is reported in Section 3. In Section 4, the differences between the three scenarios results are analysed highlighting the importance of reproducing the tidal dynamics in the MS. Finally in Section 5, the conclusive remarks.

10

# 2 The study site

The Messina Strait (figures 1a and 1b) is a narrow and deep channel connecting two Mediterranean sub-basins: the Tyrrhenian and the Ionian Sea. The channel is 70 km long and 10 km wide with the narrowest passage of about 3 km width. The main axis is north-south oriented slightly bending eastward in correspondence of the northern opening. The water depth

5 at the mouths of the strait varies between 500 m and 600 m, abruptly decreasing to 100 m in correspondence of a sill in the narrowest passage.