# Peer review of "Hydrodynamic modeling of coastal seas: the role of tidal dynamics in the Messina Strait, Western Mediterranean Sea"

_Natural Hazards and Earth System Sciences, 2016_

## Referee Comment (RC1) · Anonymous Referee #1 · 15 Apr 2016

The manuscript of the article "Hydrodynamic modeling of coastal seas: the role of tidal dynamics in the Messina Strait, Western Mediterranean Sea" by Cucco et al., presents an interesting case study on how tides contribute to the alteration of the circulation regimes at narrow straits and the connected seas. Although case-specific, the study's conclusions apply by extension to relevant modelling attempts and are deemed to present potential permanent value for literature regarding operational oceanography modelling systems.

The content of this work falls within the scopes of the Journal. The manuscript is well-structured, succeeding into not turning its length to a disadvantage; the use of English being at a high level certainly helps towards that. Materials and methods are

adequately presented; results are comprehensible and clearly laid out; discussion and conclusions are coherent to the presented results.

My recommendation is to accept the manuscript for publication in NHESS pending a few minor revisions, as noted in the following comments.

[Content]

-Given the different vertical discretization schemes of TSCRM and SHYFEM (sigma levels vs zeta levels, respectively), it would be worth commenting on how any eventual discrepancies during the model nesting were overcome.

-The authors could elaborate a bit more on the "linear combination" (mentioned in Page 7 / Line 30) that resulted into the water level open boundary conditions.

-Is there any suggestion as to why Station 10 - Messina appears to give the highest relative discrepancies in observed and computed harmonics' amplitudes?

[Presentation]

-General remark: There appears to be a general issue with the manuscript page-scaling (this also indicated by the positioning of the page numbers), which probably occurred during the discussion paper's production. This has resulted into some of the Figures not being clearly legible (especially those in which the identification of circulation patterns is needed). I would expect this to be resolved during the final production of the paper.

-General remark: In-text references to Figures and Tables are usually made with these words' first letters in capital; the same applies to the beginning of Figure/Table captions.

-General remark: The phrase "in correspondence of" is repeated throughout the text; "in correspondence to" would be grammatically correct. However, and since the meaning of the phrase with "to" does not seem to fit, I am under the impression that this could be a misinterpretation of a phrase in the authors' native language that needs to

be revised.

-General remark: The discussion on several of the Figures presenting the model's results includes references to the locations of the stations and/or the sections presented in Fig. 1. It would be beneficial for the readers if notations about these locations were included in the respective Figures as well.

-Page 1 / Lines 11-14: This sentence could be rephrased; the repetition of connecting words and the absence of punctuation make it difficult to comprehend at first.

-Page 2 / Line 3: "literature" seems more proper; consider revising.

-Page 2 / Line 19: "water current, salinity and temperature" could be replaced by "aforementioned" in order to avoid repetition; consider revising.

-Page 2 / Line 25: "based on the use" could be replaced by "using" in order to avoid repetition; consider revising.

-Page 2 / Lines 33-34: The last sentence should be rephrased (maybe adding "...are presented").

-Page 4 / Caption of Fig.1: There is a double reference to "panel b"; the second one should probably be "panel c".

-Page 5 / Line 13: "in high detail" seems more proper; consider revising.

-Page 7 / Line 4: "extent" seems more proper; consider revising.

-Page 8 / Line 4: "Results and Discussion".

-Page 8 / Line 14: "was" or "computations were".

-Page 15 / Caption of Fig.4: "Vertically" seems more proper; consider revising.

-Page 16 / Line 3: "who" seems more proper (instead of "which"); consider revising.

-Page 15 / Lines 28-30: The first sentence needs rephrasing (and/or punctuation), as

it is difficult to comprehend at first.

-Page 20 / Line 10: "we" could be omitted; consider revising.

-Page 20 / Line 30: There's a reference to "blue" lines in Figs. 7a and 7b, but - at least at the discussion paper - these lines appear as magenta; revision/adaptation needed.

-Page 23 / Line 10: Should it be "...along section AB"?

---

## Referee Comment (RC2) · Anonymous Referee #2 · 15 Apr 2016

The paper "Hydrodynamic modeling of coastal seas: the role of tidal dynamics in the Messina Strait, Western Mediterranean Sea" by Cucco et al., investigating the tidal impact on the exchange flow in the strait Messina, using numerical model techniques, is interesting and important for understanding/modeling the regional dynamics. The results are important and can be applied in operational oceanographic applications but also in longer-term (e.g. climatic) studies. The main problem with the manuscript is that it includes too many focal points (tidal validation and analysis, tidal variability impact on the exchange fluxes and tracer studies), which is reducing its coherence. I believe that the paper should be published, after minor revision, but I also think that there should be a reduction of the manuscript, in order to make it more focused and useful.

More specifically:

Figure 2 (and its description in page 10) does not provide a complete picture of the tidal flow in the region and does not contribute in the overall discussion since the paper is not focusing on identifying the tidal signal. The results of TDO are only compared to observations/previous work only for validation purposes.

Section 4.2.2, on the thermohaline features, needs more discussion on the processes involved in modifying the fluxes and T/S fields, due to tidal flow variability.

I believe that the Section 4.2.3 does not contribute to the overall focus points of the paper and does not add significant information to the results presented in previous sections and should be omitted. If the authors want to connect this work with operational needs (e.g. advection/diffusion of tracer), an operational setup of the modeling system should be considered.

Other specific comments:

How do you define the length of the strait?

Are the full slip conditions appropriate for the regional dynamics?

The tidal model validation seems incomplete (important constituents are missing). The authors should explain where data are available and where bibliographic values are used. The authors should also provide an overall estimation of the error and some indication for the accepted level of accuracy for the model.

Figure 4 caption: What do you mean by "across and along section A-B"

You should define winter and summer period used, explain the reason why you are using only 45 days for estimating the mean seasonal characteristics. Annual average values for the fluxes should also be presented and discussed.

Figure 7: it is better to use dates instead of day numbers.

[Figure]

Figure 7: Where is the blue line?

The estimation method and the reason for using the "average negative and positive fluxes" should be commented. There seems to be confusion in what you are using for creating Table 3.

Figure 8: Use latitude instead of km.

"South to/north to": should be corrected to" south of/north of" (everywhere)?

I think you should reconsider the words "correspondence" and "promoting" (everywhere in the text).

Page 26 – line 11: The phrase is wrong ("The heat budget through section AB").

Very little can be seen in figure 10. It would be better if you plot the difference.

The computation and interpretation of the ETTS should be mentioned, although reference is provided, in order to help the reader follow the discussion in this section.

---

## Author Comment (AC1) · 29 Apr 2016

Dear Editor

The paper has been modified following the reviewers suggestions and comments. Several modifications were made to improve the clarity of the text. The English grammar was revised. The general structure of the paper was slightly changed following the recommendations of the reviewers. The quality of the figures was improved and figure 2 was deleted as requested by reviewer #2.

In the following we pointed out the answers to the main comments and questions of the reviewer #1.

[Figure]

ANSWERS TO REVIEWER #1

Reviewer #1 main comment: "Given the different vertical discretization schemes of TSCRM and SHYFEM (sigma levels vs zeta levels, respectively), it would be worth commenting on how any eventual discrepancies during the model nesting were overcome."

The nesting procedure, already tested in previous works (Cucco et al., 2012a; 2012b; Melaku et al., 2015), allowed to force the SHYFEM model domain, based on unstructured mesh, with Open Boundary Conditions (OBC) provided by the TSCRM model, based on structured mesh. The OBC was built for T, S and Eta at all the nodes of the two Open Boundaries (OB). As an example, for each node of the OB, the T was derived at each z-layer of the SHYFEM model mesh. A linear interpolation was used to derive, for each node of the TSCRM, the vertical distribution of the T values at each z-layer of the SHYFEM model domain. Subsequently, by means of Lagrange interpolation procedure, the T was computed for each node of the SHYFEM model mesh starting from the new horizontal distribution of the TSCRM T calculated at each SHYFEM model z-layer. The obtained interpolated 3D fields was used both as OBC and for the nudging procedure. The adopted interpolated data was not generating any distortion in the computed SHYFEM hydrodynamic fields being the OB located off-shore and with the same geometrical and bathymetrical features of the TSCRM raw mesh. We avoided to describe the adopted method in the text being it already described in Cucco et al., (2012a; 2012b) and Canu et al., 2014.

Reviewer #1 main comment: "The authors could elaborate a bit more on the "linear combination" (mentioned in Page 7 / Line 30) that resulted into the water level open boundary conditions."

It was a mistake, the TTC water levels were derived by simple summation of the tidal and THO water level signals. Sorry for the inattention.

In the text, page 7 at lines 1-4: "Similarly, for the TTC scenario, the same ocean and

meteorological data were used as open boundary conditions with the exception of the water levels which, in this case, were derived for each point of the mesh by the sum of the tidal elevations (adopted in TDO scenario) and the sea surface elevations computed by TSCRM (adopted in THO scenario)."

Reviewer #1 main comment: "Is there any suggestion as to why Station 10 - Messina appears to give the highest relative discrepancies in observed and computed harmonics' amplitudes?"

The differences between modelled and measured semidiurnal amplitudes at Messina station are mainly due to the discrepancies between the simulated and the observed positions of the Anphidromic Point (AP). In particular, considering the values obtained for stations 4 and 7 (Ganzirri and Faro), north of the supposed AP position, an underestimation of the observed semidiurnal amplitudes are found (see table 2). On the contrary, at station 10 (Messina), south of the AP position, the modelled data overestimate the observed amplitudes (see table 2). It follows that the modelled iso amplitudes and the AP position of the main semidiurnal waves are slightly shifted to the north with respect to their supposed locations. Therefore, the model inaccuracy is not properly on the reproduction of the amplitudes of the semidiurnal tidal waves but on reproducing the exact position of the AP. This, of course, is not particularly affecting the reproduction of the tidal dynamic the Strait.

Reviewer #1 main comment: "General remark: . . ."

We followed all indications provided by the reviewer: the stations numbers have been included into the figures, "in correspondence of" has bee substituted with proper propositions, Figures and Tables have bee edited without capital letters and the figures have been uploaded with high details and resolution.

Smaller comments have been integrated silently into the manuscript

2016.

---

## Author Comment (AC2) · 29 Apr 2016

Dear Editor

The paper has been modified following the reviewers suggestions and comments. Several modifications were made to improve the clarity of the text. The English grammar was revised. The general structure of the paper was slightly changed following the recommendations of the reviewers. The quality of the figures was improved and figure 2 was deleted as requested by reviewer #2.

In the following we pointed out the answers to the main comments and questions of the reviewer #2.

[Figure]

ANSWERS TO REVIEWER #2

Reviewer #2 main comment: "Figure 2 (and its description in page 10) does not provide a complete picture of the tidal flow in the region and does not contribute in the overall discussion since the paper is not focusing on identifying the tidal signal. The results of TDO are only compared to observations/previous work only for validation purposes."

We modified the text and deleted figure 2.

Reviewer #2 main comment: "Section 4.2.2, on the thermohaline features, needs more discussion on the processes involved in modifying the fluxes and T/S fields, due to tidal flow variability. "

We agree with the reviewer that the increase in the discussion of the results is always better. Nevertheless the focus of the paper is the determination and quantification of the role of modeling the tidal dynamics in modifying the reproduction of the general circulation in the Strait, and not the analysis of the general circulation itself. In this latter case, a more robust evaluation of the seasonal variability of the TS would be desirable and necessary, but, for the specific objective of the paper, we considered the obtained results as satisfactory. Furthermore, due to the scarcity of TS data in the Strait and in the surrounding coastal zone, further analyses and insights of the model results could lead only to speculations.

Reviewer #2 main comment: "I believe that the Section 4.2.3 does not contribute to the overall focus points of the paper and does not add significant information to the results presented in previous sections and should be omitted. If the authors want to connect this work with operational needs (e.g. advection/diffusion of tracer), an operational setup of the modeling system should be considered."

We partially agree with the reviewer. The added value due to the computation of the ETTS and the consequent role of the tides in modifying it, is negligible and probably redundant. That is why we deleted all the contents related to the ETTS. On the

other hand, we believe that the analyses on the transport processes in the MS and the approach we followed to identify the differences between the scenarios, are very interesting and decisive for the scope of this study. In fact, the obtained results provide an intuitive overview of the "errors" that are done when skipping the tidal contribution from the modeling of the MS general circulation. As an example, the analyses of the transport results, considering the temporal integration of the tracer concentration provide an alternative evaluation of the impact of tidal contribution to the computation of the general circulation in the area Furthermore, the differences between the two scenarios results as obtained from hyndcast simulations are absolutely representative of the ones we will have from operational run. In fact, in both forecasting and hindcasting modes, the model setup and the boundary conditions are the same, technically speaking, and, there is no reason why the role of the tides should be different in the two cases.

Reviewer #2 main comment: "How do you define the length of the strait?."

The Strait is comprised between 37.9°- 38.3° North and 15.3°-15.8° East, considering the hydrological area with its Southern border aligned with Capo dell'Armi (Calabria) and Capo d' Ali (Sicily) and its Northern Border with Capo Rasocolmo (Sicily) and the Calabria coastline. We considered the area included by the set of sections used by Defant (1940) for his hydrological and morphological analyses.

In the text, page 3, line 5: " The Strait is comprised approximately between 37.9°- 38.3° North and 15.3°-15.8° East, it is about 40 km long, and around 10 km wide with the narrowest passage of about 3 km width."

Reviewer #2 main comment: "Are the full slip conditions appropriate for the regional dynamics?"

The full slip conditions were considered as adequate being the lowest size of the mesh on the domain borders around 50 m. For these spatial scale, in the case the element depth is shallow, the wall effects are negligible if compared to the reduction of the flow induced by the bottom friction. On the contrary, for deeper elements the wall effect is

not a realistic forcing being, in SHYFEM, the water depth treated as constant for all the element surface. The no sleep condition is more appropriate in the case the mesh size resolution is adequate and in the case the objective of the study is focused on the small scale dynamics generated by the horizontal diffusion of the momentum, which is not the case of this work.

Reviewer #2 main comment: "The tidal model validation seems incomplete (important constituents are missing). The authors should explain where data are available and where bibliographic values are used. The authors should also provide an overall estimation of the error and some indication for the accepted level of accuracy for the model."

We considered the 7 major diurnal and semidiurnal components of the tides in the Mediterranean Sea as stated by many authors and adopted in numerous studies (as an example, see all the applications of SHYFEM model to the Adriatic Sea and Venice lagoon, https://sites.google.com/site/shyfem/list-of-publications ). Regarding the compound tides and the shallow water tides, we considered only the M4 wave, being the most intense and its amplitudes and phases well documented by direct observations. The source of observed amplitudes and phases for each station and for each component is well documented in the text.

In the text at page 7, line 20: " The main harmonics observed amplitudes and phases were estimated by Vercelli, (1925) during the homonym Oceanographic cruise in 1922, and more recently reported by Brandolini et al., (1980) and Androsov et al., (2002a). In this work we refer to the values from Brandolini et al., (1980) for stations from 1 to 6 and from Androsov et al., (2002a) for stations from 7 to 10."

In the text, the table 2 allow a direct comparison between model results and observation for each tidal wave and for each station. In the text at page 7-8 the model accuracy in reproducing the major semidiurnal, diurnal and compound tide is discussed. We also estimated the overall accuracy of the model results considering the whole set of

stations and all the tidal components.

In the text at page 8 line 19: " On average, considering the whole set of stations and tidal components, the model results reproduced the observed amplitudes with a RMSE of 0.78 cm which is fair and acceptable for this type of analysis."

Reviewer #2 main comment: "You should define winter and summer period used, explain the reason why you are using only 45 days for estimating the mean seasonal characteristics."

We clarified the method followed to analyze the model results.

In the text at page 10, lines 16: "In figures 4 and 5 the residual currents computed for winter and summer seasons as the algebraic averages of the three-dimensional hourly current speeds obtained for winter (January, February and March) and summer months (July, August and September) during the 2 years of simulation are reported for both scenarios."

In the text at page 11, lines 21-22:" Water fluxes through section CD were computed during both winter and summer months from the THO and TTC scenarios results. The winter period included January, February and March, whereas the summer period included the months of July, August and September of both simulated years. As an example, in figure 6 the time series of the computed fluxes are reported, for reason of clarity, only for 45 days during both summer and winter 2015 and for the THO and TTC scenarios. "

Reviewer #2 other comments

Figure 7 now includes the dates.

Caption of table 3 now describes clearly all the contents.

In the text, caption table 3:" Table 3: Water, salt and heat fluxes through section CD computed for winter and summer periods from THO and TTC scenario results and dif-
ferences (TTC-THO). Water and salt fluxes are expressed in Sv, heat fluxes in 107 Watt. The South Flux and North Flux are considered as absolute values. The Net Flux (N-S) values are obtained as the differences between North Flux and South Flux values. Positive values indicate a northward net flux, negative values indicate a southward net flux. "

Smaller comments have been integrated silently into the manuscript